# Compound Characterization of a *Mucuna* Seed Extract: L-Dopa, Arginine, Stizolamine, and Some Fructooligosaccharides

Ana Lilia Hernández-Orihuela [1], Karla Viridiana Castro-Cerritos [2], Mercedes Guadalupe López [3] and Agustino Martínez-Antonio [4,*]

1   Biofab México, 5 de Mayo 517, Irapuato 36500, Mexico
2   Centro de Investigaciones Científicas, Instituto de Química Aplicada, Campus Tuxtepec, Universidad del Papaloapan, Circuito Central 200, Tuxtepec 68301, Mexico
3   Departamento de Biotecnología y Bioquímica, Centro de Investigación y de Estudios Avanzados del IPN-Unidad Irapuato, Irapuato 36824, Mexico
4   Laboratorio de Ingeniería Biológica, Departamento de Ingeniería Genética, Centro de Investigación y de Estudios Avanzados del IPN-Unidad Irapuato, Irapuato 36824, Mexico
*   Correspondence: agustino.martinez@cinvestav.mx; Tel.: +52-4626269600

**Abstract:** Human societies demand sustainable alternatives for goods and services. Plants are sustainable sources of important metabolites with beneficial impacts on human health. There are many reported methodologies and commercial suppliers for extract preparations from *Mucuna* sp. They usually claim the plant is enriched with L-dopa, its distinctive metabolite. However, many present poor characterizations of the extract's components. Here, we present polar metabolites characterization of a *Mucuna* seed extract, emphasizing L-dopa identification and quantification. To obtain the extracts, we follow a green and sustainable extraction protocol. The lyophilized extract is subjected to liquid chromatography and mass spectrometry to identify its primary metabolites. Additionally, we follow thin-layer chromatography to identify carbohydrates in the sample. The resultant extract has 56% L-dopa. Other main components in the extract are arginine, stizolamine, and the fructooligosaccharides sucrose and nystose. The characterized *Mucuna* extract can be easily standardized using powder preparation and used in several biomedical applications.

**Keywords:** *Mucuna* seed; L-dopa content; lyophilized extract; sugars identification; TLC (thin layer chromatography); mass spectrometry





## 1. Introduction

The development of human societies has been linked to using plants for food, materials, and medicines [1]. One example of the use of these plants is the legume velvet bean, also known as *Mucuna* sp. [2]. The origins of this plant can be traced to areas of China, Malaysia, and India [3]. Accessions of these plants are now present in tropical regions [4], including the southeast of México [5]. In Central America, farmers use the plant in culture rotation with milpa to recover and improve soil nutrition [6], as green manure, and as a cover crop [7], among related uses. Some other uses of *Mucuna* seeds are as feed ingredients in poultry nutrition [8]. The seed shell has also been used for wastewater paint treatment [9]. Additionally, methanolic extract from the leaves of this plant shows broad antimicrobial activity [10]. The most ancient practical use of *Mucuna* for improving human health is in Ayurvedic medicine [11]. The use of *Mucuna* seeds has been documented to have several pharmacological benefits. Their usefulness includes antidiabetic [12], aphrodisiac [13], antineoplastic [14], antiepileptic [15], anti-venom [16], antihypertensive [17], and anti-neurodegenerative properties [18], and the improvement of male fertility [19], among others. These applications are due to their unique metabolite composition [6]. A typical application of *Mucuna* seed is to alleviate Parkinson's disease [20]. This beneficial effect relies on L-dopa (levodopa) activity (3,4-Dihydroxyphenylalanine), mainly present in this plant's

seeds [21]. Levodopa is a precursor of the neurotransmitter dopamine, but is unable to cross the blood–brain barrier. As a result, L-dopa is administered to patients with damaged dopaminergic neurons, the treatment by choice for Parkinson's disease [22]. Unfortunately, conform progresses the condition, and after years of therapy, L-dopa becomes less effective and provokes some complications such as dyskinesia [23]. Carbipoda, an inhibitor of dopamine carboxylase, is dispensed together with L-dopa to avoid the rapid enzymatic degradation of dopamine [24].

Whole beans, mainly toasted, are consumed to alleviate or prevent Parkinson's disease [25]. However, treating Parkinson's with seeds is difficult as it requires high seed doses since the content of L-dopa in the grain is maximally reported to be 9.16% [26]. In addition, some people present adverse effects to consuming the seeds, such as vomiting. One alternative is to concentrate the L-dopa in *Mucuna* seed extracts [27], which can be more manageable and dried as a powder where L-dopa is demonstrated to be stable [28]. The whole seeds have shown advantages in patients with Parkinson's [29] and animal models versus synthetic L-dopa [30]. Thus, using seed extracts could improve the management of dose quantity. L-dopa in seed extracts is adequate to alleviate Parkinson's disease at 6 mg/kg [31]. Another study calculated an equivalent daily levodopa dose of 100 mg [32]. The selective extraction of L-dopa relies on the chemical interactions between L-dopa and the extractant solvent. The process conditions diminish the extraction of most metabolites in the seed yet do not avoid the coextraction of some metabolites, which maintain the advantages of these preparations over synthetic L-dopa [33]. Many protocols have been developed to extract L-dopa from *Mucuna* seeds; these range from using water [34] and alcohol, but most of them employ hydro-acid solutions [27]. Some are assisted by microwave or supercritical $CO_2$ [35].

However, there are no characterizations of the molecules in these extracts beyond L-dopa, and their direct use in animal experiments is customarily carried out [30,31,33]. Notably, there are many commercial suppliers of both seeds in powder and seed extracts with a wide range of levodopa declared content in their labels. In a study with six commercial products, the authors found that the labels indicate levodopa doses from 60 to 250 mg L-dopa/dose. Still, standardized HPLC quantification found variations from 14 to 705 mg L-dopa/dose, with a deviation corresponding from 6 to 141% concerning the claim on the product label [36]. In another study with 15 products of the NIH Dietary Supplement Label Database, using UHPLC, the authors found that L-dopa varies enormously in these products [37]. In addition, these products are poorly characterized, and no metabolite composition is reported. This information scarcity makes their appropriate use and the attribution of potential benefits to specific molecules difficult.

In this study, we use a Mexican accession of *Mucuna pruriens* sp. to obtain a lyophilized extract enriched with L-dopa that can be used in biomedical applications. We identify and quantify L-dopa and the sample's other significantly related components.

## 2. Materials and Methods

### 2.1. Seed Material, Their Production, and L-Dopa Extract Preparation

Seeds of *M. pruriens* var. *ceniza* (Figure 1) were kindly gifted by Prof. Castillo Caamal [38]. The seeds were cultivated organically on farmland fertilized with cattle manure without chemicals or pesticides in the community of Tepecoacuilco, State of Guerrero, Mexico (18°18′0″ N, 99°29′0″ W). Cultivation largely depends on the rain-based season, typically from June to January. The harvest of pods occurred when they turned from green to dark. Pods were collected from plants and dried in the sun to liberate the seeds. The clean kernels were stored in containers at room temperature and protected from direct light until their use. This work presents the average results of seeds collected in 2020 and 2021.

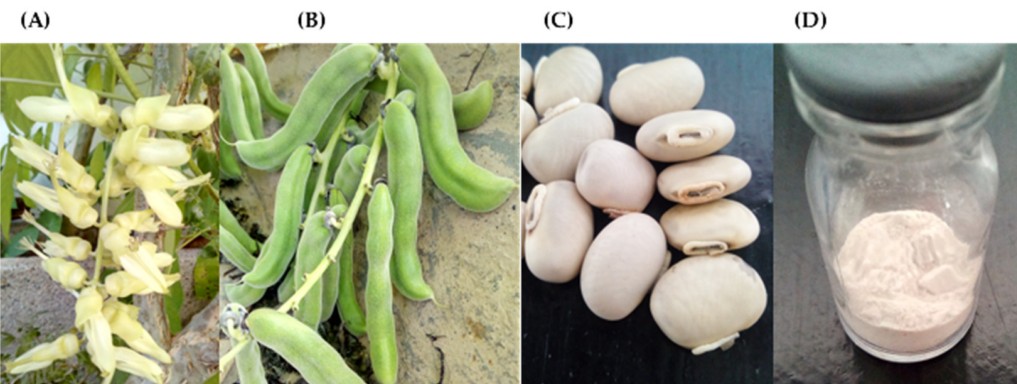

**Figure 1.** *Mucuna* pruriens sp. and seed extract. Different organs and physiological states of the plant are shown: flowers (**A**), green pods (**B**), mature seeds (**C**), and lyophilized powder extracted from seeds (**D**).

There are many reported methods for L-dopa extractions from *Mucuna* [27]. Given that we were interested in implementing an organic and sustainable process, we mostly followed the extraction method reported by Polanowska et al. [39]. We slightly modified the protocol by dissolving the equivalent of 100 g milled dried seed in 1 L of 0.3% acetic acid and 0.1% citric acid (1:10 *w/v*). Then, we followed two extractions steps for 20 min each at room temperature. After clearing by centrifugation at 10,000 rpm for 10 min (OHAUS, Frontier 5816R, Parsippany, NJ, USA), the supernatant was freeze-dried in a Virtis Freezemobile 12 lyophilizer (ALT Inc., East Lyme, CT, USA). The lyophilized powder was stored in a dark container at 4 °C; we identified this extract product as lot BFLD21-003.

*2.2. Sample Preparation for Polar Metabolites Analysis*

In total, 5.0 mg of the freeze-dried powder were dissolved in 1.0 mL of 1% formic acid and centrifuged at 10,000 rpm for 5 min to eliminate any insoluble material. The supernatant was passed through an SPE C18 cartridge (3 mL Sep-Pak[R], Waters, Milford, MA, USA). This column was pre-activated with 1.0 mL of methanol and equilibrated with 1.0 mL of 1% formic acid. Then, the sample (1.0 mL) was passed, and the eluates were collected. Next, 450 μL of the eluate were taken, and 50 μL of methanol were added to reach a solvent composition similar to the mobile phase in the column used in Section 2.4 (below) and to obtain reproducibility regarding the retention times. This final solution was filtered using a 0.22 μm PTFE membrane (Millex[R], Merck, Darmstadt, Germany), and 1.0 μL was injected into the chromatographic system for metabolite analysis.

*2.3. Sample Preparation for L-Dopa Quantification*

To estimate the quantity of L-dopa in the powder extract, we first took 5.0 mg of the freeze-dried powder and dissolved it in 1.0 mL methanol with 1% formic acid. Then, it was passed through an SPE C18 cartridge (Sep-Pak[R], Waters, Milford, MA, USA). The cartridge was pre-activated with 1.0 mL of 50% methanol and equilibrated with 1.0 mL of 1% formic acid. Then, the sample was passed by the cartridge and washed with 1.0 mL methanol 40%. Both eluates (flowthrough and wash, about 2 mL) were collected and diluted at 50 mL with formic acid 1% to reach a methanol concentration similar to the mobile phase. In total, 200 μL of this final dilution were filtered using a 0.22 μm PTFE membrane (Merck, Darmstadt, Germany), and 0.1 μL was injected into the chromatographic system for L-dopa analysis.

*2.4. Analysis of Polar Metabolites by Liquid Chromatography Coupled to Mass Spectrometry (UPLC-ESI-TOF-MS)*

Polar metabolites analysis of the extract was carried out in a liquid chromatograph (Acquity UPLC I-Class, Waters, Milford, MA, USA). The chromatography equipment was coupled to a high-definition mass spectrometer (Synapt G2-Si, Waters, Milford, MA, USA).

This was equipped with an electro-nebulization ionization source, a single quadrupole mass filter, an ion mobility system, a collision cell, and a time-of-flight mass analyzer (ESI-Q-SIM-CID-TOF). We operated the LC-MS system and the analysis and the spectral data acquisition with the Masslynx 4.1 program (Waters, Milford, MA, USA).

Chromatographic separation was performed with a column Luna $^®$ Omega C18 100 Å, 1.6 μm (150 × 2.1 mm, Phenomenex, Torrance, CA, USA). The composition of the mobile phases was 0.1% formic acid and methanol in a 49:1 ratio using a flow of 200 μL/min at 40 °C and run in an isocratic mode.

For metabolites analysis, samples were carried out in both positive and negative ionization modes. The mass spectrometer was operated with the following parameters: capillary voltage, 3000 V; cone voltage, 40 V; source temperature, 120 °C; cone nitrogen flow, 50 L/h; nebulizer nitrogen pressure, 6.5 bar; nitrogen temperature for desolvation, 350 °C; nitrogen flow for nebulization, 800 L/h. The mass spectra were acquired every second in a continuous format. The spectral range *m/z* was set from 50 to 1200. Two spectral functions were captured for each LC-MS analysis using argon as a collision gas, low energy at 6 V, and an energy gradient from 20 to 60 V. Spectral correction was carried out by continuously infusing the reference compound leucine-enkephalin: 556.2771 for ESI positive and 554.2615 for negative ESI. The mass spectrometer was calibrated using NaI (Waters, Waters, Milford, MA, USA). Spectral analysis was carried out with the Progenesis Q.I. (Waters, Milford, MA, USA) program and was verified manually. The compounds identified with a retention time of 1.52 and 3.15 min were confirmed with L-arginine and L-dopa standards, respectively (CAS 74-79-3 and CAS 59-92-7. Sigma Co., Burlington, MA, USA). No commercial standard is available for stizolamine; therefore, we identified it by MS spectrum only.

For L-dopa quantification, the mass spectrometer was operated in ESI positive mode, and the mass spectra were acquired every 0.4 s in a centroid format. L-dopa was monitored at *m/z* 198.076 ± 0.01 Da.

### 2.5. Estimating the Quantity of L-Dopa in the Extract

First, we developed a calibration curve from 50 to 500 pg of commercial L-dopa (Sigma Co., Burlington, MA, USA) to estimate the quantity of L-dopa in the powder extract. We followed the chromatographic method reported in ref. [40] by calculating the area of each calibration point in the mentioned liquid chromatography coupled to mass spectrometry equipment. The standard error (S.E.) and standard deviation (S.D.) were calculated using linear regression. The limit of detection (LOD) and limit of quantification (LOQ) were calculated using the formulas LOD = 3.3 ∗ (SD/SE) and LOQ = (10 ∗ SD/SE).

We extrapolated the quantity of L-dopa in the 1.0 μL sample as prepared in Section 2.3 with the calibration curve and its linear equation. L-dopa concentration in the powder was calculated considering the 1:50 dilution achieved in Section 2.3.

### 2.6. Thin Layer Chromatography (TLC) Analysis

For TLC analysis, 10 mg of the lyophilized powder extract were dissolved in 90 μL of DMSO (Sigma Co., Burlington, MA, USA) and 20 μL of 1N HCl. FOS (fructooligosaccharides) and MOS (maltooligosaccharides) standards were purchased from Sigma Co. One and two μL of either sample extract or commercial standards were applied to silica gel TLC plates with aluminum support (10 cm × 10 cm, Merck, Darmstadt, Germany). TLC plates were developed three times with a mobile phase made of butanol/propanol/water (3:12:4, *v/v/v*). TLC was carried out three times: one was visualized with U.V. light, another developed with ninhydrin to derivatize amino acids such as compounds (L-dopa), and another with a solution of aniline/diphenylamine/phosphoric acid reagent in the acetone base to reveal carbohydrates [41].

## 3. Results

### 3.1. Obtention of the Lyophilized Extract

Figure 1 shows the different organs of the *Mucuna* plant from which seeds were collected over two years (2020 and 2021). We carried out green L-dopa extraction following the green protocol reported in ref. [39]. We obtained a final average quantity of 101 g of lyophilized powder starting from 500 g of dried, milled seeds (Figure 1C). This quantity represents a mass recovery of 20% (*w/w*) in the extract concerning the seeds (Table 1). The final lyophilized powder was white with a slightly yellow appearance (Figure 1D). This lyophilized powder was used in the analytic approaches described below.

**Table 1.** Resume of two lot seeds corresponding to the 2020 and 2021 harvest to obtain L-dopa extracts.

| Initial Seed Mass (g) | Lyophilized Extract Mass (g) | L-Dopa in Extract (%) | Yield (Average of Two Lot Extracts) | Estimated L-Dopa in Seeds (%) |
|---|---|---|---|---|
| 500 | 101 | 56 | 20.2 | 11.20 |

To calculate the different parameters, we followed ref. [40].

### 3.2. Metabolomic Analysis of M. Pruriens Extracts by Liquid Chromatography Coupled to Mass Spectrometry (UPLC-ESI-TOF-MS)

3.2.1. Monitoring of Metabolites in Positive Ionization Mode (ESI+)

The polar fraction of the plant extract was separated using solid phase extraction (see Methods section), and the analysis of the metabolites was achieved using LC-MS equipment. In the positive ionization mode, we observed three prominent peaks corresponding to retention times of 1.52, 3.15, and 6.36 min (Figure 2). The retention time and mass spectrum of the 1.52 min peak correspond to the molecule of arginine where the primary ion $[M+H]^+$ with an *m/z* 175.1196 corresponds to the protonated form of the amino acid arginine ($C_6H_{14}N_4O_2$) (Figure 3).

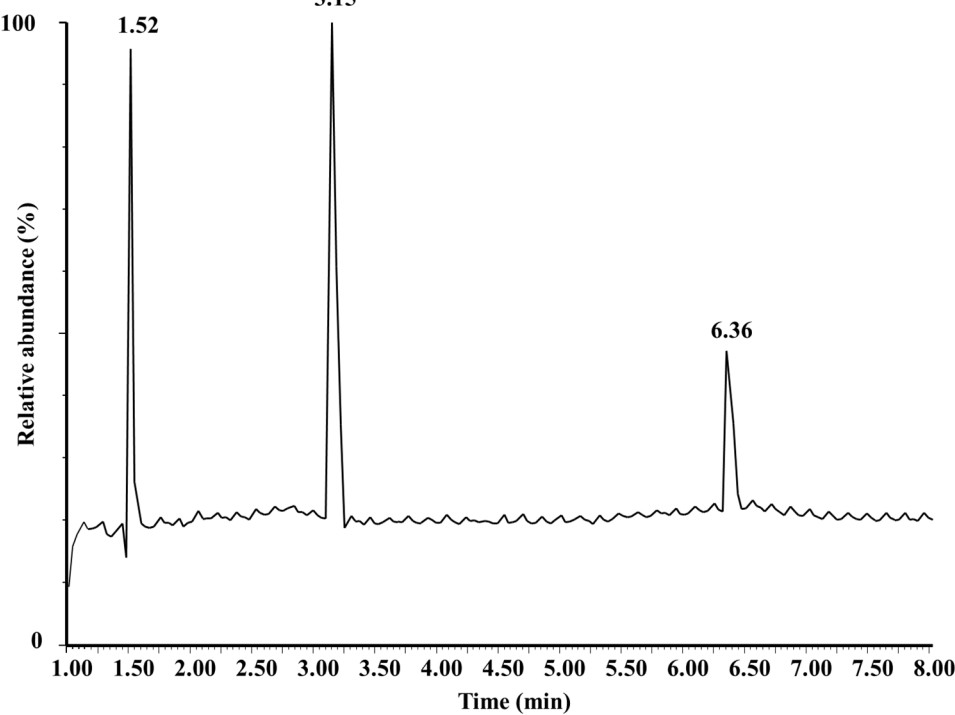

**Figure 2.** Base peak intensity chromatograms (BPI) of *Mucuna pruriens* extracts in LC-MS in positive ionization mode (ESI+).

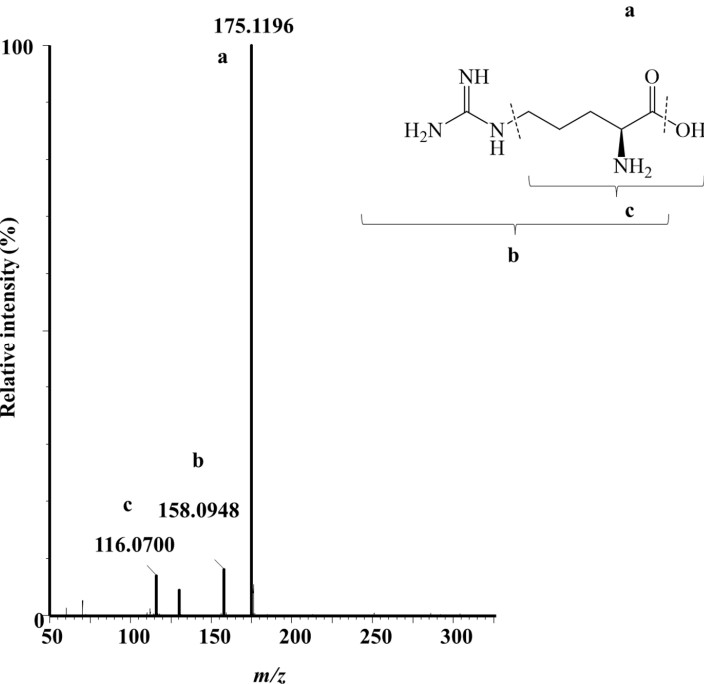

**Figure 3.** Mass spectrum and proposed structure of the compound at 1.52 min (arginine). The complete molecular ion (a) when there is a loss of the hydroxyl group (b) and when there is a loss of three terminal nitrogen atoms (c).

Furthermore, Figure 4 shows the mass spectrum of the peak corresponding to the retention time of 3.15 min in Figure 2. The mass spectrum shows an ion with an *m/z* of 198.0788 (Figure 4a) corresponding to the protonated form of the molecule $C_9H_{11}NO_4$, identified as L-dopa. We verified this putative identity by comparing its retention time and mass spectrum with a commercial standard of L-dopa (Sigma Co., Burlington, MA, USA). The most abundant ion *(m/z* 152.0694) corresponds to the L-dopa molecule with a loss of its carboxyl group. Additional fragmentation peaks are observed where different functional groups of the molecule are lost (Figure 4).

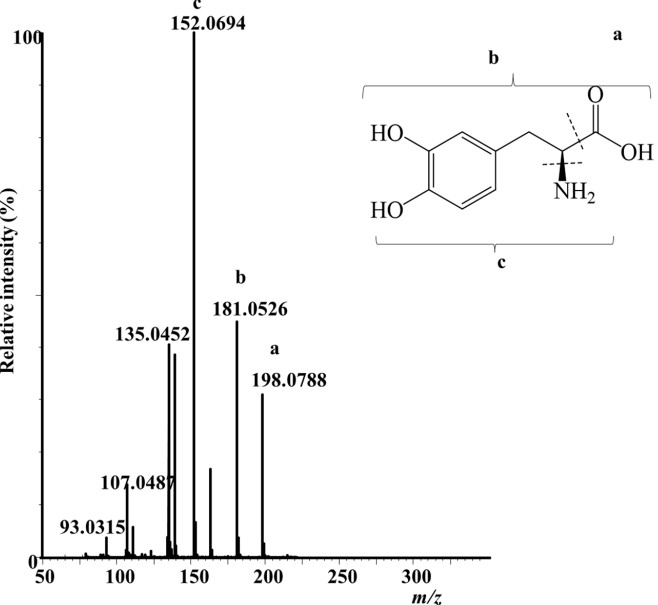

**Figure 4.** Mass spectrum of L-dopa (peak 3.15 min in Figure 2). The parent ion (*m/z* 198.0788) (a) when an amino group (*m/z* 181.0526) (b) or carboxyl group (*m/z* 152.0694) (c) is lost.

Figure 5 shows the mass spectrum of the compound with a retention time of 6.36 min in Figure 2. It reveals a primary ion with an *m/z* of 198.0972. This ion was identified as the protonated form of a compound with the molecular formula $C_7H_{11}N_5O_2$, which corresponds to stizolamine. However, we cannot confirm the molecule's identity since there is no standard commercially available.

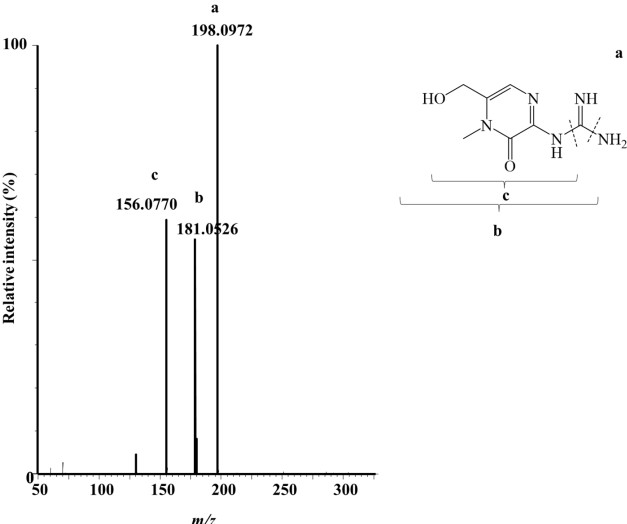

**Figure 5.** Mass spectrum and proposed structure of stizolamine. The complete molecular ion (a) and when one and two amino groups generating the ions with *m/z* 181.05 (b) and m/z 156.07 (c) are lost, respectively.

### 3.2.2. Monitoring of Metabolites in Negative Ionization Mode (ESI−)

We also analyzed the extract in negative ionization mode using liquid chromatography coupled with mass spectrometry (UPLC-ESI-TOF-MS). Figure 5 shows the most abundant peak compounds in this analysis. The general profile of metabolites results was very different concerning the positive mode. Two significant peaks appeared during negative ionization mode (Figure 6).

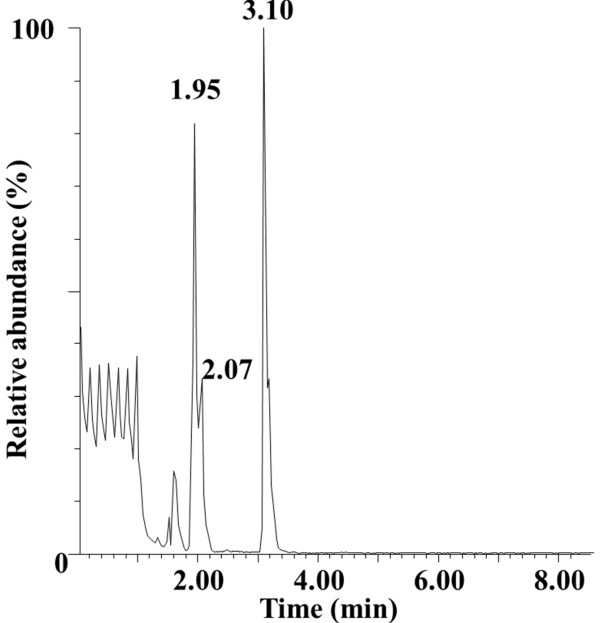

**Figure 6.** Base peak chromatogram of *Mucuna pruriens* extracts in LC-MS in negative ionization mode (ESI−). Two peaks appear majorly.

Peaks with retention times 1.95 and 2.07 most possibly correspond to the coelution of several oligosaccharides with the molecular formulas $C_{30}H_{52}O_{26}$ (pentamer), $C_{24}H_{42}O_{21}$ (tetramer), $C_{18}H_{32}O_{16}$ (trimer), and $C_{12}H_{22}O_{11}$ (dimer), as revealed by their respective mass spectra (Figure 7).

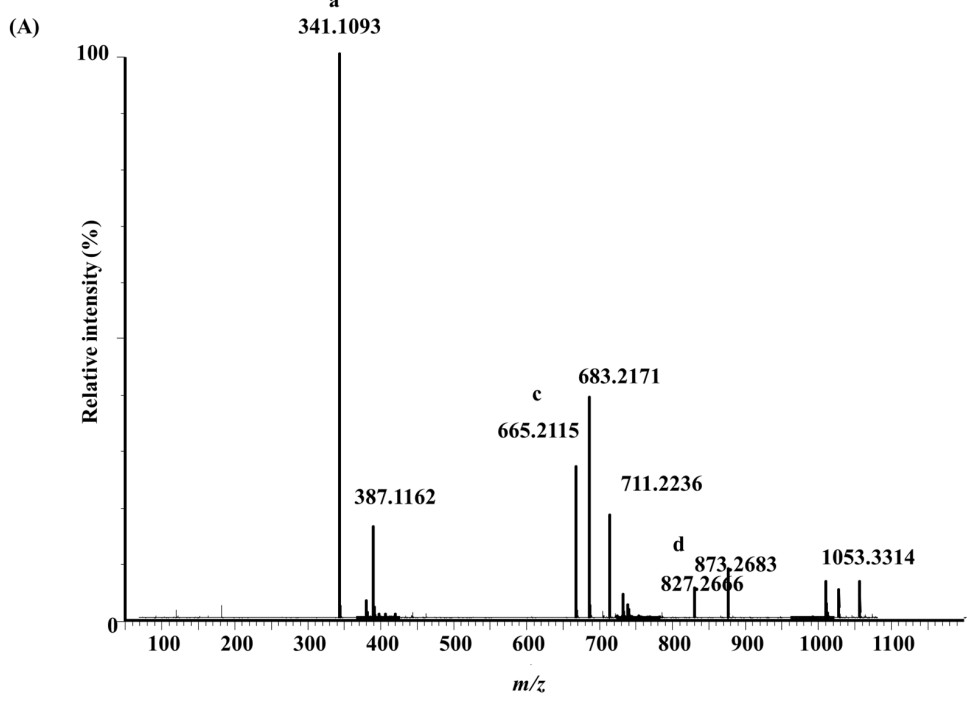

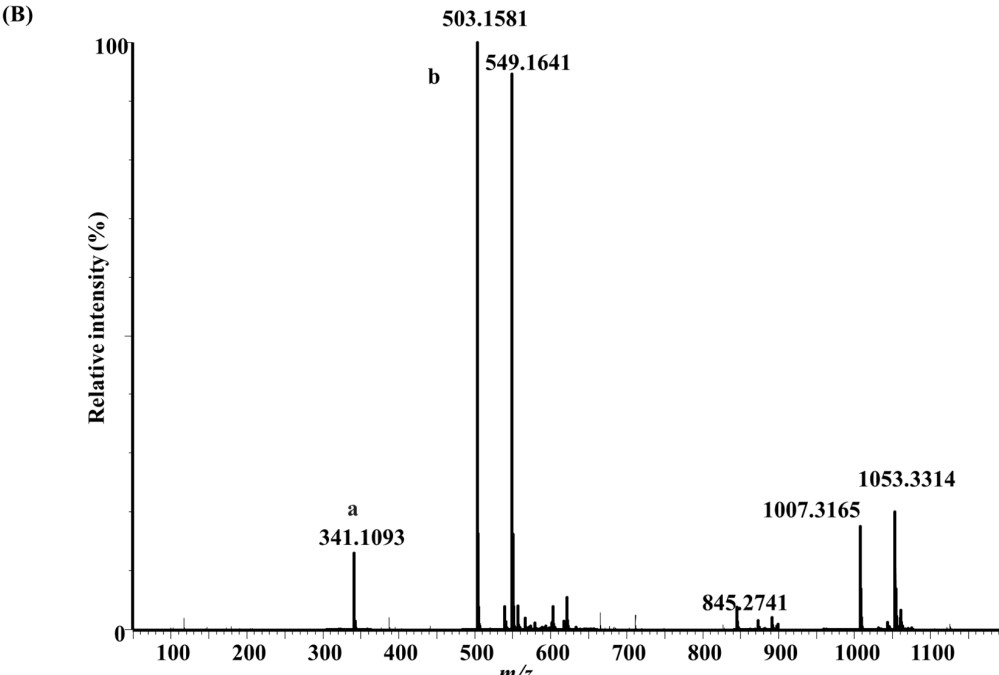

**Figure 7.** Mass spectra of the peaks with retention times 1.95 (**A**) and 2.07 (**B**), as in Figure 6, monitored in ESI (−). The deprotonated form $[M-H]^-$ of dimeric sugar $C_{12}H_{22}O_{11}$ (*m/z* 341.11), trimer $C_{18}H_{32}O_{16}$ (*m/z* 503.15), tetramer $C_{24}H_{42}O_{21}$ (*m/z* 665.21), and pentamer $C_{30}H_{52}O_{26}$ (*m/z* 827.26) are depicted by (a–d), respectively. The oligomers form ionic adducts in the presence of formic acid, resulting in ions with an increase of +46 *m/z* (387.12, 549.16, 683.21, and 873.27).

Finally, the peak in retention time of 3.10 min in Figure 6 corresponds to the deprotonated form [M−H]⁻ of L-dopa (Figure 8).

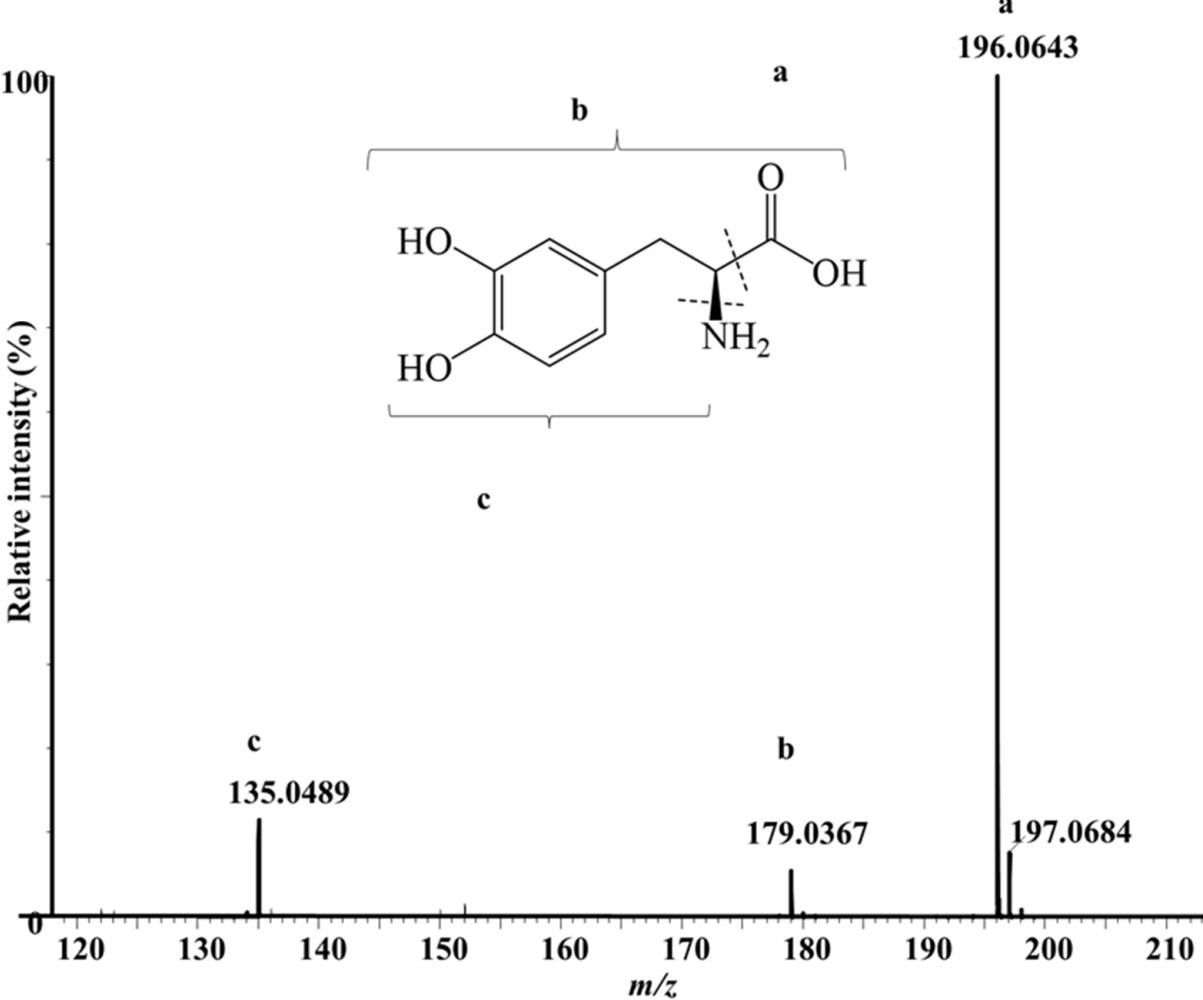

**Figure 8.** Mass spectrum of L-dopa in negative ionization mode. (a) Parent ion (*m/z* 196.0643), (b) loss of amino group (*m/z* 179.0367), and (c) loss of amino and carboxyl group (*m/z* 135.0489) are depicted in the image.

### 3.3. Quantification of L-Dopa in the Mucuna Seed Extract

To quantify the presence of L-dopa in the extract, we developed a calibration curve with a commercial standard of L-dopa. In Figure 9A, we can observe a point of the calibration curve corresponding to 5000 pg of standard L-dopa. Figure 9B shows the calibration curve for L-dopa. Using the calibration curve, we can calculate that we have 0.56 mg per mg extract, which corresponds to a 56%, which means a little more than half of the lyophilized mass in the extract corresponds to L-dopa.

### 3.4. Thin Layer Chromatography (TLC) Identifies the Presence of Fructooligosaccharides in the Mucuna Seed Extract

Chromatographic analysis of the TLC shows that the sample presents a tiny spot when photographed under normal light (line 2 in Figure 10A). A coincident spot in the upper part of the sample is shared with the commercial standard for L-dopa (line 1). When the sample was exposed to U.V. light, it displayed a strong blue fluorescence, as mentioned in the case of stizolamine. However, no previous image is available (line 2, Figure 10B). This fluorescence was not present in the standard as expected. Ninhydrin, on the other hand (Figure 10C), allowed the localization of amino acids such as L-dopa in the sample,

as in the standard. Finally, when the standard of several maltooligosaccharides (MOS) was run together with the sample, it revealed the presence of oligosaccharides but not like MOS (Figure 10D). When a standard of fructooligosaccharides (FOS) was compared, in an independent run from A–D, the first spot on Figure 10E, line 3 corresponded to sucrose, and there was a second coincident spot corresponding to 1-nystose (N-DP4). However, the sample's coloration was different, possibly corresponding to one sugar modification. The commercial standard of L-dopa did not present oligosaccharides as expected.

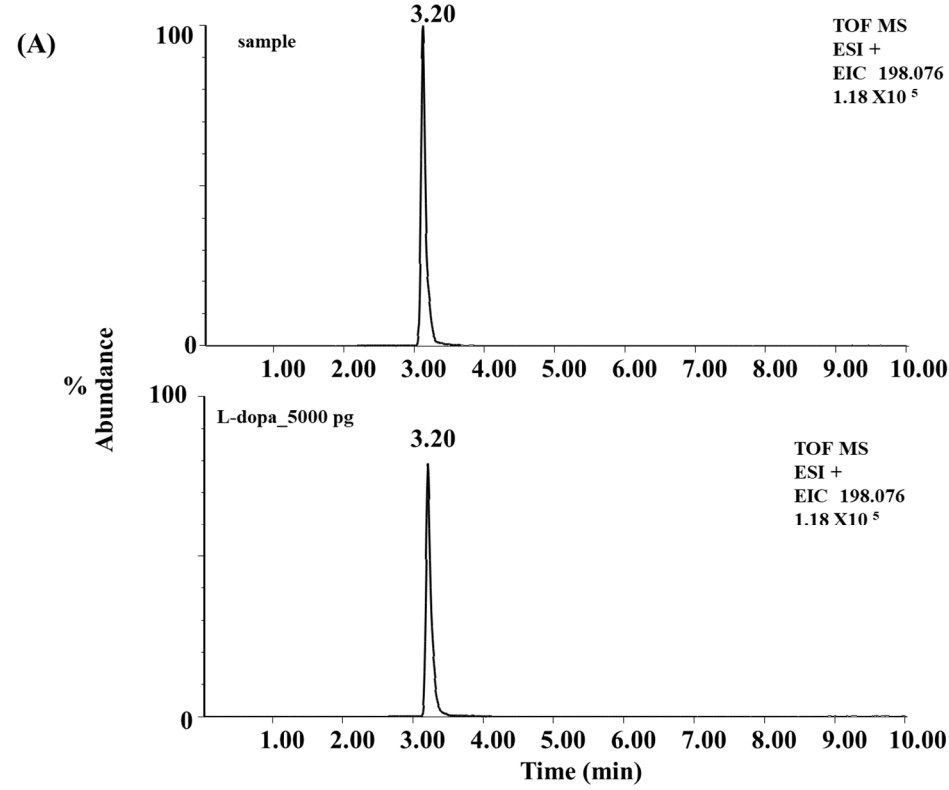

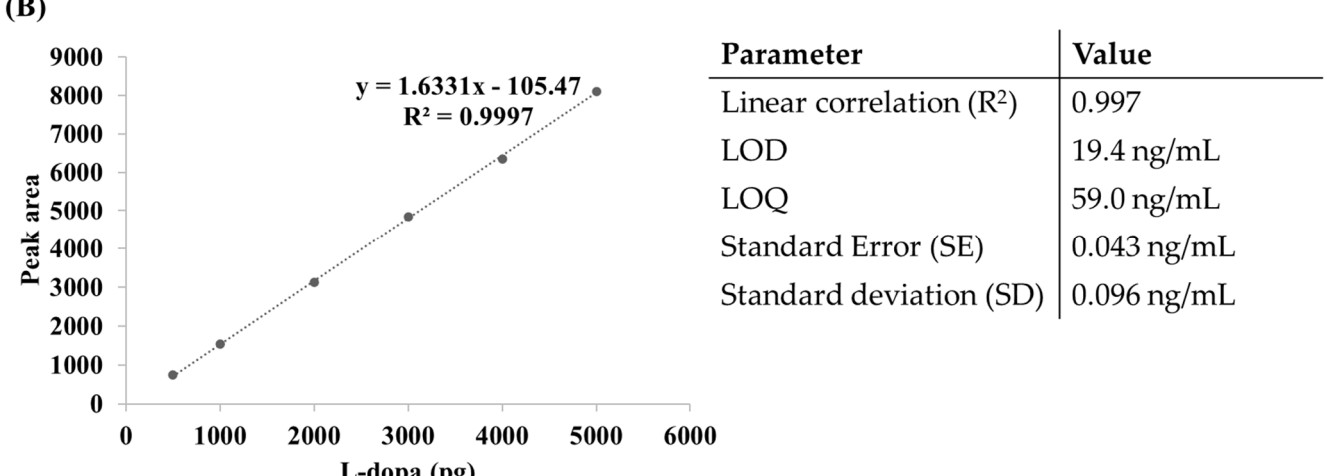

**Figure 9.** Quantification of L-dopa. (**A**) Extracted ion chromatograms for $m/z$ 198.076 of the sample (**up**) and 5000 pg of standard L-dopa (**down**). (**B**) Calibration curve with the commercial L-dopa with their respective statistical parameters (**right**).

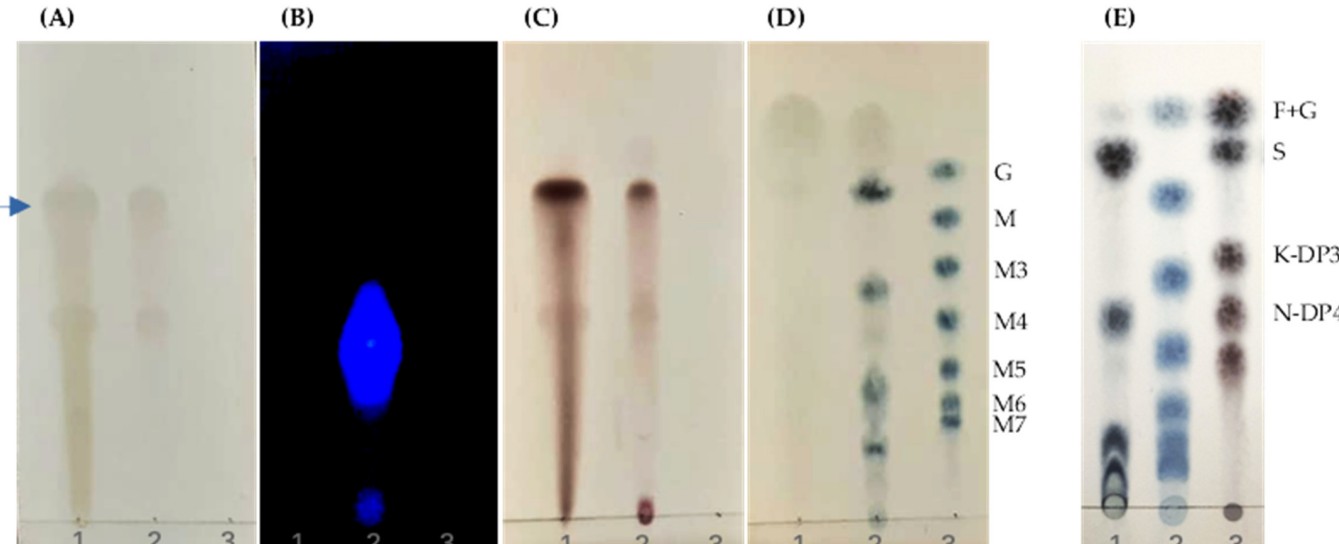

**Figure 10.** Thin layer chromatography of the *Mucuna* seed extract. In the same run (**A**–**D**), line 1 corresponds to 2 μL of L-dopa standard, line 2 to 1 μL of the extracted sample, and line 3 to 1 μL of maltooligosaccharides (MOS) standard mixture. (**A**) Revealed under normal light, (**B**) under U.V. light, (**C**) with ninhydrin, and (**D**) with aniline/diphenylamine/phosphoric acid. (**E**) An independent run revealed as in (**D**), line 1 corresponds to 2 μL of the sample; line 2 corresponds to 1 μL of maltooligosaccharides (MOS) and line 3 corresponds to 1 μL of fructooligosaccharides (FOS) standard mixture. MOS; G, glucose; M, maltose; M3, maltotriose; M4, maltotetraose; M5, maltopentose; M6, maltohexose; M7, maltoheptaose. FOS; F+G, fructose plus glucose; S, sucrose; K, 1-kestose (DP3); N, 1-nystose (DP4); FFN, 1-fructofuranosyl-nystose (DP5).

## 4. Discussion

*Mucuna* seed extracts are commercially available for human consumption as food supplements and are widely used in many experiments in animal models. These experimental approaches look for evidence of the different qualities of the plants. Despite this, in most of these studies, there is no characterization of the extracts. There is only L-dopa quantification, although their content does not always correspond with the declared content label of commercial products [37]. In this work, we decided to investigate the component resultants on a *Mucuna* seed extract to be more precise on the potential effects it could have during experimentations. Previous investigations report the feasibility of recovery yields and validation of methodology working with L-dopa [42,43]. We were sure our extraction methodology was directed to extract L-dopa, but inevitably, we co-extracted additional compounds. Therefore, our first approach was looking for L-dopa with a technique that permits us to differentiate it from other molecules. In this way, we started by analyzing metabolites using liquid chromatography coupled with mass spectrometry (UPLC), followed by the ionization of metabolites by electrospray (ESI), hitting the detector in a way that discriminated ions of the same *m/z* value with different initial energies (TOF-MS). These analyses were conducted in both positive and negative ionization modes. The positive ionization mode revealed three well-defined peaks, and their further fragmentation revealed the amino acid arginine, L-dopa, and stizolamine. Our quantification of L-dopa confirms we have an appropriate extraction method since a little more than half of the mass weight in the sample corresponded to L-dopa (56% *w/w*). We extrapolated a content of 11.2% of L-dopa in the source seeds. Several plants can be natural sources of L-dopa; *Mucuna* pruriens are reported as those with major content of L-dopa with an utmost of 9.16% [26]. Our methodological approach may result in overquantifying L-dopa in seeds since our results were higher than the literature reported.

We found in old literature that in addition to L-dopa, stizolamine was also reported in *Stizolobium* [44,45]. *Stizolobium* is synonymous with *Mucuna* of the Fabaceae plant

family [46]. It is important to emphasize that L-dopa and stizolamine are molecules already registered as present in *Mucuna* plants [47]. A straightforward way to differentiate stizolamine from L-dopa is that the former is said to present strong blue fluorescence under U.V. light, although this has not been pictured. In our case, we confirmed the presence of a compound with blue fluorescence in our sample. Additionally, this was not present in the standard, as shown in Figure 9B. Stizolamine pertains to the class of compounds of pyrazines, has biological functions such as pollinator pheromones [48], and pyrazines are of some putative medical importance (e.g., anticancer and antidiabetic) [49]. The third abundant component in the positive mode was arginine, although it eluted in the no retained fraction of the column and masked additional components. We confirmed that the peak corresponds to commercial standards, though this was not the methodological approach appropriate for their quantification. If further confirmed, would be intriguing to determine the highest quantity of this amino acid since it is not the most abundant amino acid reported in *Mucuna* seeds (Table 2).

The literature reported a structural analog to arginine, canavanine, in Canavalia-related plants [50]. Arg-tRNA can charge canavanine instead arginine. This substitution disrupts the metabolism as a defensive way for this plant to deter predators [51]. Both molecules (arginine and canavanine) present very relative molecular weights, differing in just two mass units. Deep analysis shows us that the equipment already detects canavanine, but just at the noise level. Therefore, we are sure that we visualized arginine. Arginine is the most basic charged amino acid, and the methodological process could favor its extraction. Arginine is encoded by six codons and is found in high quantities in dietary nuts [52]. Arginine was also the most abundant amino acid among the principal storage proteins of the megagametophyte of the Loblolly pine tree. It was hypothesized as an essential compound for nutrition and seedling growth [53]. In addition to glutamine, arginine has been proposed as a crucial immune nutrient for improving health [54]. In a metastudy with 10 randomized control trials, the authors concluded that supplements with arginine, in doses of 1.5–5 g/day, could be recommended for mild to severe erectile dysfunction [55]. As a result, arginine can promote male fertility.

**Table 2.** Reported content of amino acids in *Mucuna* pruriens seeds.

| Compound/Reference | | | | | | | |
|---|---|---|---|---|---|---|---|
| **Amino Acids (g/100 g Protein)** | **Ref. [56]** | **Ref. [57]** | **Ref. [58]** | **Ref. [59]** | **Ref. [60]** | **Ref. [61]** | **Ref. [62]** |
| Glutamic acid | 17.23 | 12.87 | 10.8 | 19.31 | 12.3 | 10.38 | 9.80 |
| Aspartic acid | 8.16 | 17.71 | 11.2 | 17.10 | 11.4 | 13.11 | 13.40 |
| Serine | 4.10 | 14.33 | 5.6 | 6.08 | 4.37 | 4.24 | 4.20 |
| Threonine | 3.64 | 8.60 | 5.1 | 5.21 | 3.58 | 3.56 | 4.21 |
| Proline | ND | NR | 7.2 | 7.38 | 5.06 | 2.44 | 2.14 |
| Alanine | 2.81 | 4.09 | 6.5 | 4.95 | 3.22 | 5.12 | 4.80 |
| Glycine | 5.12 | 4.98 | 8.1 | 6.21 | 4.30 | 4.90 | 4.95 |
| Valine | 5.57 | 2.94 | 5.8 | 7.60 | 4.23 | 3.56 | 2.98 |
| Cysteine | 0.84 | 4.30 | 0.4 | 1.61 | 1.01 | 1.01 | 1.72 |
| Methionine | 1.28 | 0.00 | 1.1 | 0.78 | 0.72 | 0.74 | 0.79 |
| Isoleucine | 4.12 | 4.27 | 5.2 | 8.77 | 4.16 | 6.24 | 6.40 |
| Leucine | 7.85 | 2.43 | 7.3 | 10.42 | 7.88 | 5.94 | 6.78 |
| Tyrosine | 4.76 | 3.84 | 2.8 | 7.51 | 4.45 | 4.24 | 4.52 |
| Phenylalanine | 3.85 | 0.80 | 4.5 | 6.51 | 4.70 | 4.01 | 3.63 |

**Table 2.** *Cont.*

| Compound/Reference | | | | | | | |
| --- | --- | --- | --- | --- | --- | --- | --- |
| **Amino Acids (g/100 g Protein)** | **Ref. [56]** | **Ref. [57]** | **Ref. [58]** | **Ref. [59]** | **Ref. [60]** | **Ref. [61]** | **Ref. [62]** |
| Tryptophan | 1.35 | NR | NR | ND | 1.22 | 0.56 | 0.94 |
| Lysine | 6.60 | 7.17 | 6.0 | 8.98 | 6.18 | 6.61 | 6.80 |
| Histidine | 3.14 | 5.22 | 2.6 | 3.30 | 3.47 | 2.90 | 2.43 |
| Arginine | 7.16 | 5.28 | 4.5 | 9.55 | 5.28 | 6.66 | 6.80 |

ND, not detected; NR, not reported.

Another important fraction component in *Mucuna* seed extract, revealed in negative mode, is the mixture of oligosaccharides. UPLC-ESI-TOF-MS analysis in negative ionization mode presented a peak at 1.95, whose mass spectrum predicted the dimer $C_{12}H_{22}O_{11}$, which probably corresponds to sucrose or cellobiose; TLC analysis confirmed that it was sucrose (a dimer consisting of glucose and fructose). MS analysis predicted higher oligomers, but by TLC, we ensured $C_{24}H_{42}O_{21}$ corresponded to 1-nystose, consisting of glucose and three fructose. TLC analysis revealed the presence of at least two higher fructose oligomers, but we could not identify them with our standards (Figure 10D, line 1 bottom). Further characterization is required to ensure the identity of these complex oligosaccharides, although the precise identification of oligosaccharides is not common for herbal supplements. Alternatively, further purification should be sought to eliminate these oligosaccharides in the sample, mainly if their ingestion results in gastrointestinal disorders.

There is scarce metabolite analysis reported for *Mucuna* extracts. GC-MS analysis of seeds methanolic extracts revealed the presence of five major compounds: Pentadecanoic acid, 14-methyl-, methyl ester, Dodecanoic acid, 9,12-Octadecadienoic acid (Z, Z)-, methyl ester, 9,12-Octadecadienoic acid, and 2-Myristynoyl-glycinamide [63]. It is possible that methanolic extraction rends more volatile compounds in this extract regarding our extracting approach. In another study, using whole seeds and a combination of chromatographic and NMR techniques, the authors reported the presence of d-chiro-inositol and its two galactose derivatives in *Mucuna pruriens* [64]. Inositol and galactose derivatives may have been present in our seeds but were diluted in our seed extract.

With this study, we aimed to contribute to the molecular characterization of an extract of this important plant. Our metabolic approach mainly focused on studying chemically polar metabolites such as L-dopa. Therefore, no polar metabolites were retained in the column, and different methods are necessary to confirm or discard the presence of additional metabolites in relevant quantities. This would provide a more complete description of the botanical product so more information is available for consumers and health professionals. Deep knowledge of the components in the product is essential because there are many commercial preparations of *Mucuna* with poor or no ingredient characterization. There is a remarkable opportunity successfully chemically characterize phytochemical products so we can be more confident of their attributed health properties.

## 5. Conclusions

Here, we reported on the simple preparation of a lyophilized *Mucuna* seed extract enriched with 56% of L-dopa (*w/w*). This quantity represents the average of two harvests of seeds in different years (2020 and 2021). To our knowledge, this is an academic report with a higher concentration of L-dopa for seed extract. In addition to L-dopa, arginine, stizolamine, and some fructooligosaccharides were also present in the extract. Neither of these components reported primary risk effects on health. Further studies are required to explore the existence of additional metabolites with different chemical natures in our extracts (e.g., volatiles or nonpolar). Plant extracts require standardized characterizations to achieve comparable studies and profitable medical applications.

**Author Contributions:** Conceptualization, K.V.C.-C., M.G.L. and A.M.-A.; methodology, A.L.H.-O., K.V.C.-C. and M.G.L. validation, A.L.H.-O., K.V.C.-C. and M.G.L. formal analysis, K.V.C.-C., M.G.L. and A.M.-A.; investigation, K.V.C.-C., M.G.L. and A.M.-A.; resources, K.V.C.-C., M.G.L. and A.L.H.-O.; writing—original draft preparation, A.M.-A.; writing—review and editing, K.V.C.-C., M.G.L. and A.M.-A.; project administration, A.L.H.-O.; funding acquisition, A.L.H.-O. All authors have read and agreed to the published version of the manuscript.

**Funding:** This research and APC were funded by BIOFAB MEXICO, grant number DP2021-1. The funder had no inference in the design and conclusions of the investigation. LC-MS facilities were granted by CONACYT INFRA-2015-01-252013 and 317105.

**Institutional Review Board Statement:** Not applicable.

**Informed Consent Statement:** Not applicable.

**Data Availability Statement:** Not applicable.

**Acknowledgments:** The authors thank José Bernardino Castillo Caamal for providing us with the *Mucuna* seeds. We also thank Miguel Angel Ramos Valdovinos, Danae Carrillo Ocampo, Verónica Fabela Garatachía, Berenice Cuevas, and Tomás Tiburcio for their technical contributions.

**Conflicts of Interest:** The authors declare no conflict of interest. The funder had no role in the design of the study, in the collection, analysis, or interpretation of data; in the writing of the manuscript; or in the decision to publish the results.

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
