# Peer review of "Compound Characterization of a Mucuna Seed Extract: L-Dopa, Arginine, Stizolamine, and Some Fructooligosaccharides"

_compounds, doi:10.3390/compounds3010001_

Round 1
Reviewer 1 Report
Comments to authors:
The authors submitted the manuscript entitled "Compounds characterization of a Mucuna Seed Extract: L-dopa, arginine, stizolamine, and some fructooligosaccharides" to be able to publish at Compounds.
The authors use a Mexican accession of Mucuna pruriens sp. to extract and lyophilize a L-dopa from Mucuna Seed for several biomedical applications. In fact, the work is very interesting but the authors need to be presented the experiment and results in a very clear way.
The manuscript still needs to be revised and improved in order to be able for publication consideration. So that I cannot recommend the publication of the manuscript in its actual state.A major revisionis recommended.
Some comments are below:
- In the abstract, the authors need to be presented the % values of other mentioned components such as arginine and stizolamine.
- The introduction part should be covered all the important of extracts.
- The authors mentioned that they followed a green and sustainable extraction protocol but they didn’t explain in the extraction procedure.
- The conclusion is very short and need to be improved.
Author Response
Reviewer 1
The authors submitted the manuscript entitled "Compounds characterization of a Mucuna Seed Extract: L-dopa, arginine, stizolamine, and some fructooligosaccharides" to be able to publish at Compounds.
The authors use a Mexican accession of Mucuna pruriens sp. to extract and lyophilize a L-dopa from Mucuna Seed for several biomedical applications. In fact, the work is very interesting but the authors need to be presented the experiment and results in a very clear way.
The manuscript still needs to be revised and improved in order to be able for publication consideration. So that I cannot recommend the publication of the manuscript in its actual state.A major revisionis recommended.
Some comments are below:
- In the abstract, the authors need to be presented the % values of other mentioned components such as arginine and stizolamine.
R. Thank the reviewer, as explained in the responses to the other reviewers, we cannot give approximate values in these compounds. First, our metabolic approach is not the most correct for quantifying arginine, and second, there is no commercial standard for stizolamine.
- The introduction part should be covered all the important of extracts.
R. Thank you, in different parts of the manuscript, we refer to the importance of plant extracts in human use. At the discussion and conclusion, we mention the importance of having standardized characterizations of products based on plant extract to have comparable studies and applications.
- The authors mentioned that they followed a green and sustainable extraction protocol but they didn't explain in the extraction procedure.
R. Thank you. In this revised version, we describe the extraction process followed in more detail.
- The conclusion is very short and need to be improved.
R. We rewrote the conclusions and mentioned the important points derived from this study. Thank you for your observations.
Reviewer 2 Report
The paper “Compounds characterization of a Mucuna Seed Extract: L-dopa, arginine, stizolamine, and some fructooligosaccharides” by A.L. Hernández-Orihuela et al. provides a metabolite characterization of Mucuna seed extract. It is a typical herbal supplement, and in these types of products, it is necessary to characterize them before commercialization.
The paper is well-written and presents a simple analytical procedure for this characterization, and this is always positive. However, my main concern is about the recovery test not being present (you have included a lyophilization step not present in the original procedure, although studied) and the need to assay more than a single sample.
2.1. “Seeds of six cultivars …” “Of these, M. pruriens var. ceniza …” It is not clear from the document whether this specific variety is the only one present or whether a mixture of different varieties has been used. In the latter case, you have to study the results for the six cultivars and check if they are similar.
“2.2 Sample preparation for metabolite quantification” The sample is dissolved in 1 mL methanol (1% formic acid), but then 10 mL are passed by the column. Something is not well described in the middle.
What is the reason for the 50 microL addition of methanol to the 450 microL of the sample?
“2.4 Analysis of metabolites by LC…” It is not an important issue, but why do you use acetic acid here and not formic acid? (as in the preparation step)
“3.3 Quantification of L-dopa in the Mucuna seed extract” In addition to the calibration line, a table with all the relevant statistics of it (number of points, uncertainties for the intercept and slope, LOD, LOQ, …) must be presented.
“4. Discussion”
“Our quantification of L-dopa confirms we have an appropriate extraction method since a little more than half of the mass weight in the sample corresponds to L-dopa (56% w/w).” How does this value compare with other bibliographic results?
“Alternatively, further purification is required to eliminate these oligosaccharides in the sample.” Do you think this is a necessary step in an herbal supplement?
The paper is of potential interest, but it is necessary:
- Present a recovery study about the amount of L-dopa transferred from the sample to the extract (for example, by using fortified samples).
- Present the results for more samples. If you have seeds from six different cultivars you should have six different results.
Author Response
Reviewer 2
The paper "Compounds characterization of a Mucuna Seed Extract: L-dopa, arginine, stizolamine, and some fructooligosaccharides" by A.L. Hernández-Orihuela et al. provides a metabolite characterization of Mucuna seed extract. It is a typical herbal supplement, and in these types of products, it is necessary to characterize them before commercialization.
R: Thank you, the reviewer, for your kind consideration of our work.
The paper is well-written and presents a simple analytical procedure for this characterization, and this is always positive. However, my main concern is about the recovery test not being present (you have included a lyophilization step not present in the original procedure, although studied) and the need to assay more than a single sample.
2.1. "Seeds of six cultivars …" "Of these, M. pruriens var. ceniza …" It is not clear from the document whether this specific variety is the only one present or whether a mixture of different varieties has been used. In the latter case, you have to study the results for the six cultivars and check if they are similar.
R: We are sorry for the confusion it could generate in our writing. We are located in the middle of Mexico, and the people that gifted us the seeds are in the southwest. Then the seed most adapted to grow in our location was Mucuna pruriens. That is why we follow our characterization only with this specie and not the other five (by the way, the production of L-dopa was almost the same on the six plant accessions). To help to clarify this point, we have now eliminated the reference to the other accessions in the manuscript and just left Mucuna pruriens.
"2.2 Sample preparation for metabolite quantification" The sample is dissolved in 1 mL methanol (1% formic acid), but then 10 mL are passed by the column. Something is not well described in the middle.
R: thank you for the observation. You are right. Now we correct it to 1.0 mL in both cases.
2.3 What is the reason for the 50 microL addition of methanol to the 450 microL of the sample?
R: We added methanol to the sample to equal the solvent conditions favoring the polar metabolites in the sample, like the mobile phase in the column. This detail was now introduced in the text.
"2.4 Analysis of metabolites by LC…" It is not an important issue, but why do you use acetic acid here and not formic acid? (as in the preparation step)
R. Sorry, we corrected acetic to formic acid (as it was used).
"3.3 Quantification of L-dopa in the Mucuna seed extract" In addition to the calibration line, a table with all the relevant statistics of it (number of points, uncertainties for the intercept and slope, LOD, LOQ, …) must be presented.
R. Thank you. We were unsure if adding this information. Now it is included as part of figure 9 and further describes their methodological part in section 2.3.
"4. Discussion"
"Our quantification of L-dopa confirms we have an appropriate extraction method since a little more than half of the mass weight in the sample corresponds to L-dopa (56% w/w)." How does this value compare with other bibliographic results?
"Alternatively, further purification is required to eliminate these oligosaccharides in the sample." Do you think this is a necessary step in an herbal supplement?
R. Indeed, it is not necessary to quantify oligosaccharides in an herbal supplement. We state this might not be necessary.
The paper is of potential interest, but it is necessary:
- Present a recovery study about the amount of L-dopa transferred from the sample to the extract (for example, by using fortified samples).
R. Thank you. We are sure this is a great suggestion and will be doing it for putative applications of the extract added to the fortified sample to ensure the absolute quantity of L-dopa in that preparation. Some basic information about L-dopa in seeds and the extract was included in Table 1.
- Present the results for more samples. If you have seeds from six different cultivars you should have six different results.
R. Thank you. However, we now clarify that we are using just a single cultivar because it is the most adapted to secure production. We think your suggestion is excellent to follow the annual production to ensure that differences are in each annual production because we have extraction replicas of just two productions corresponding to 2020 and 2021.
Thank reviewer 2; their comments greatly help to improve our manuscript.
Reviewer 3 Report
Abstract
1. „poor characterizations of the metabolite’s components”
Should read “extract’s components.”
2. “standardized as powder presentation”
In the abstract and throughout the manuscript the word “presentation” is used. Do the authors refer to a pharmaceutical “preparation”?
Introduction
3. “One example of the use of these plants is the legume velvet bean, also known as Mucuna spp. [2].”
This is outside my area of expertise, but as far as I know the legume known as velvet bean consists of a single species (Mucuna pruriens). Why is the plural form “spp.” used?
4. “The extracting procedures are directed to obtain L-dopa based on their reacting chemical nature to solvents.”
Clarification/re-phrasing needed.
5. “However, there are few characterizations of the molecules in these extracts beyond L-dopa, and their direct use in animal experiments is customarily done [36].”
The authors claim that animal experiments are customarily done, citing [36] as reference. However, [36] does not provide information on animal exeriments. On the other hand, the previous references do provide a characterization of the molecules in these extracts beyond L-dopa, which is the opposite of what the first part of the sentence claims.
6. “The rest of the manuscript describes our methodological approach, the results obtained, and a brief discussion and conclusions.”
The usual composition of the manuscript (“methods”, “results”, “discussion”, and “conclusion”) need not be explained to the reader.
Materials and Methods
7. “we mostly followed the extraction method reported by Polanowska et al. [40].”
Even if the protocol has been published before, a brief summary containing all critical parameters such as extraction time and temperature would be useful for readers seeking to reproduce the extraction, especially since the cited reference [40] contains several extraction protocols.
8. “5.0 mg of the freeze-dried powder was dissolved in 1 ml of 1% formic acid…”
For mass, two significant figures are given (“5.0 mg”). For volume and other parameters, only one significant figure (“1 mL”). Does this imply different levels of accuracy tolerance?
9. 2.2 Sample preparation for metabolite quantification
Major comments
A) “The supernatant was passed through an SPE C18 cartridge (ThermoFisher Scientific).” Which type (e.g. “Hypersep”) and mass (e.g. 100-1000 mg) of C18 material was used?
B) The original sample was dissolved in 1 mL. Later in the same section, 10 mL of sample are passed through the SPE cartridge. Please explain if this means that samples were combined of if there was an additional dilution step not mentioned so far.
C) SPE changes the sample composition. Was the fraction that was removed from the original extract (during washing, or still retained on column) further characterized in terms of metabolite composition?
10. In “2.3 Sample preparation for L-dopa quantification,” the SPE cartridge is pre-activated (50% MeOH), equilibrated (formic acid), loaded (sample), and washed (40% MeOH). What was used to elute the sample? Was it the 50 mL of 1% formic acid mentioned later?
11. 2.5 Estimating the quantity of L-dopa in the extract
A) Was the calibration curve obtained directly from diluted standards or after work-up identical to sample preparation (i.e. SPE)?
B) Was L-dopa used as supplied from Sigma Co.? If so, why do the authors specify its synthetic origin via L-tyrosine? In order to establish enantiopurity?
Results
12. 3.2.1. Monitoring of metabolites in positive ionization mode (ESI+)
Major comments
A) What was the void time of the chromatographic system?
This time is needed to determine the capacity factor k’ of the analyte eluting at 1.52 min. If the capacity factor is very low, there may be more than one analyte hiding under this peak, even if MS is used as detector. For example, the respective FDA Reviewer Guidance (Validation of Chromatographic Methods) recommends k’ > 2 to prevent this.
B) Was the identity of the two metabolites other than L-dopa (arginine and stizolamine) verified by comparison of the retention time and spectrum of a commercial standard?
C) The authors detect only two amino acids and one amine in the positive mode in the extract of a complex biological seed sample. How does this finding (both qualitatively and quantitatively) compare to what is known from the literature about the amino acid composition of the seed? A discrepancy would suggest a high extraction selectivity, loss of analytes during the sample preparation (SPE), or undetected analytes by the employed RP-LC-MS method [which may also be related to A)].
13. The caption of Figure 3 does not contain the name of the presumed compound (arginine).
14. Figure 9
A) The chromatogram needs to be provided in a higher resolution to be able to discern minor peaks.
B) In addition to the chromatogram after injection of 5000 pg standard, the chromatogram of a representative sample would be helpful to the reader (since the chromatogram shown in Figure 2 was obtained under different extraction and LC-MS conditions).
15. 3.3 Quantification of L-dopa in the Mucuna seed extract
Major comments
With the described extraction and quantification protocol the authors find that 56% of the extract consists of L-dopa, which appears very high compared to literature reports of L-dopa content in the raw seeds. Given this finding, the following would be helpful
A) A table summarizing the reported composition of Mucuna seeds, especially regarding amino acid and amine content (%)
B) A table summarizing the L-dopa content (%) of other reported Mucuna extracts
C) A table fulfilling the minimum requirements of method validation (at least accuracy and precision of medium quality control samples (MQC) that were not used to establish the calibration curve).
Discussion
16. “although their content does not always correspond with the declared on the label [38].”
What does this mean? What label?
17. “Further characterization is needed to ensure the identity of these complex oligosaccharides.”
Although not fully elucidated, it is assure in the conclusion of the manuscript that none of the saccharides have adverse health effects. Given the possibility of adverse GIT reactions to certain saccharides, this statement may need to be rephrased.
18. “Deep knowledge of the components in the product is essential because there are many commercial presentations of Mucuna with poor or no components characterization.”
Major comments
Given the proclaimed ubiquity of commercial presentations of Mucuna, have the authors applied their methodology to available commercial preparations? This information would allow the reader to compare the high L-dopa content in the extract obtained by the authors to existing formulations.
References
19. A cursory examination revealed that at least some of the references are not correct.
Examples:
[10] DOI given (10.4314/ajb.v6i18.57964) is different from the actual DOI (10.5897/AJB2007.000-2324).
[39] No title; URL not found.
Author Response
please see the pdf attachment

Round 2
Reviewer 1 Report
The authors answer all comments in a reasonable way and modified the manuscript as suggested. So, I recommend the acceptance of the manuscript as it is.
Author Response
Thank you, your suggestions were very usefulness.
Reviewer 2 Report
The authors have improved the first version of the paper. Now it is more interesting for the reader than before.
I still miss the recovery study to guarantee the validity of the results. It could have been done in one work session, and it would provide additional evidence for the validation of the results.
Author Response
We are very sorry. Unfortunately, the main equipment broke down, and it is impossible to do this test even though it is relatively simple. We will continue working on the extract and are left with this pending task. Thank you very much for your comprehension.
Reviewer 3 Report
Due to its size, the comments have been uploaded as a separate file.
